# Conceptualising Inclusion and Participation in the Promotion of Healthy Lifestyles

**DOI:** 10.3390/ijerph19169917

**Published:** 2022-08-11

**Authors:** Richard Peter Bailey, Suria Angit

**Affiliations:** Centre for Academic Partnerships and Engagement, University of Nottingham Malaysia, Semenyih 43500, Selangor, Malaysia

**Keywords:** inclusion, promotion of healthy lifestyles, physical activity, sport, community, marginalised groups

## Abstract

The central tension in health promotion is between a public health policy presumption that healthy lifestyles have benefits at both the individual and societal levels and should be assertively encouraged, and liberal demands that states should maintain a stance of non-interference concerning private affairs. This tension is heightened when the engagement of marginalised or disempowered groups, such as persons with disabilities, women, or immigrants, enter discussions. This paper examines the concept of inclusion within the context of the promotion of healthy lifestyles, primarily sport and physical activity in community contexts. Using a form of ‘reflective equilibrium’, it explores a series of distinctions to evaluate critically different accounts of inclusion and offers a novel and somewhat radical approach based on re-interpretations and alignments of participation and hegemonic relationships.

## 1. Introduction

The central tension in health promotion is, perhaps, between a public health policy presumption that healthy lifestyles benefit at both the individual and societal levels and should be assertively encouraged, and liberal demands that states should maintain a stance of non-interference concerning private affairs [1]. Most of the literature in this area seems to be focused on salvaging a concept of individual liberty from what would seem to be an inevitable paternalism on the part of the state [2]. This line of discussion examines the legitimacy of government influence over individual’s lives and actions, from complete non-interference to the removal of choice (as happened in many countries during COVID-19 quarantines). Its popularity among researchers is understandable in light of the rather awkward questions it raises about many tenets of the dominant liberal tradition.

Focusing on the promotion of healthy lifestyles, especially health-enhancing physical activities, this article aims to examine another aspect of health promotion, namely the inclusion of community members in local-level sporting and physical activities (PA). Following the World Health Organisation Regional Committee for Europe [3], we understand healthy lifestyles to mean a way of living that lowers the risk of being seriously ill or dying early and promotes a positive state of well-being. Health-enhancing PA is activity that contributes to this goal. Aside from the inherent value of exploring a context that has received little attention within wider conceptual discussions, we hope to show some of the ways in which questions of inclusion in public health are more nuanced and fuzzier than individual/society distinctions allow.

The relationship between PA and health has now been established beyond doubt, and awareness of the health costs of sedentary behaviours is so advanced among scientists and policymakers that inactivity is now recognised as a major public health concern [4]. Worldwide, more than 1.4 billion adults do not reach the recommended levels of health-enhancing physical activity (HEPA) and are, therefore, at greater risk of developing non-communicable diseases (NCDs) such as cancer, heart disease, stroke and diabetes [5]. The global pandemic of physical inactivity [6] is responsible for more than 5 million deaths [7], and at least EUR 61.5 billion of economic burden per year [8]. The World Health Organisation [9] reported research findings that indicate a global cost of physical inactivity INT$ 54 billion in direct health care, including substantial costs to the public sector and in terms of lost productivity. Estimates from both high-income and low- and middle-income countries indicate that between 1–3% of national health care expenditures are attributable to physical inactivity. In contrast, accumulating sufficient Moderate to Vigorous Physical Activity (MVPA) is a key determinant of physical, mental, social, and environmental health [10]. Among children and youths (aged 5–17 years), several systematic reviews have reported PA benefits in terms of physical, developmental, psychological, cognitive and social health, as well as academic achievement [11].

Despite the reported benefits, there is now an extensive literature on inequalities and exclusions from health and health promotion, and much of this understandably focuses on the experiences of socially marginalised groups [12]. The analysis is typically based on avoidable and systematic differences in health between different groups of people, in which environmental influences shape individuals’ experiences and lead to inequalities in health outcomes. The mounting evidence base in this area makes the reasoning seem unarguable [13]. However, two notes of caution can be made about this approach, one general, the other related to the specific concerns of this paper. The general point is that analyses such as this risk conflating the symptoms of health problems with their causes, and in doing so pay insufficient attention to the processes—rather than the outcomes—of inequality. It is important to acknowledge this issue as measures enacted to reduce indicators of inequality, such as health, will not necessarily succeed in fully addressing a fairer distribution of resources if they fail to address the processes behind them.

Put another way, while most countries have enacted political equality and equality before the law, it does not follow that this translates to equality of opportunity or life chances, much less equality of treatment or membership in society [14]. For these reasons, we prefer to talk about inclusion/exclusion, rather than equality/inequality, as framing the discussion of health seems better able to accommodate consideration of these processes [15]. More importantly, theorising on inclusion has tended to focus more on how individuals and groups become included or excluded from society [16]. This leads to the second cause of hesitancy. The current paper is concerned with individuals’ participation in practices associated with healthy lifestyles, and cause–effect models are too broad-brush for our purposes. This is not a criticism of that approach, as its intention was different from ours. Nevertheless, questions of participation in healthy lifestyles, such as PA, healthy eating, and mental health-supporting activities, require some insight into the lived experiences of individuals and groups interacting with these contexts.

So, this article aims to examine these issues and work towards some tentative suggestions for resolving them. Using the concept of inclusion as the core of our discussion, a form of ‘Reflective Equilibrium’ [17] is used to explore and draw coherence from competing accounts. We begin by critically discussing some of the key concepts and distinctions that have shaped current conversations about inclusion. We consider ways in which some of the resources generated by existing discourses of marginalisation and exclusion can be generalised into a more thorough-going account of inclusion within the context of promoting healthy lifestyles. We conclude by offering our thoughts about the elements of an adequate theoretical framework for making sense of inclusion within the context of promoting healthy lifestyles.

### Some Initial Comments on Inclusion

Theoretical discussions of inclusion are notoriously opaque, and this is equally true within the context of health promotion [18]. On the surface, it is a simple idea: it is the opposite of exclusion. It challenges practices that mean certain groups—the disabled, minority ethnic groups, immigrants, LGBTI+ communities—are not recognised as full participants in all aspects of society and demands the removal of barriers and social structures that interfere with participation. This presentation is important for the subsequent discussion as it asserts an account of inclusion that is at once an issue of participation and also of social justice. Framing inclusion in this way connects with recent accounts that seek to extend beyond the popular equation of inclusion with the mere presence of marginalized groups and individuals in mainstream settings. These accounts assert a fuller acknowledgement of the political and ethical issues shaping the treatment of the powerful and powerless within society. It resonates with other social justice issues such as calls for women’s suffrage, civil rights, and disability rights, which share a dual goal: to decrease exclusionary pressures and to increase the participation of marginalised groups within the culture and practices of society. The former without the latter seems self-defeating. So, inclusion is inseparable from participation and the acceptance of the fact of diversity. In this respect, inclusion is comparable with democracy as both can be interpreted in different ways, but most accounts assume some sort of equitable interaction between members. This was the view of John Dewey [19]:

*“A democracy is more than a form of government; it is primarily a mode of associated living, of conjoint communicated experience. The extension in space of the number of individuals who participate in an interest so that each has to refer his own action of that of others, and to consider the action of others to give point and direction to his own, is equivalent to the breaking down of those barriers of class, race, and national territory which kept men from perceiving the full import of their activity.*”(p. 87)

Bailey [20] sought to distil and make concrete some of the most salient features of inclusion/exclusion by positing a series of connected dimensions:*Spatial*: inclusion relates to proximity and the closing of social and economic distances;*Relational*: inclusion is defined in terms of a sense of belonging and acceptance;*Functional*: inclusion relates to the enhancement of knowledge, skills and understanding; and*Power*: inclusion assumes a change in the locus of control.

We will return to these concepts later, but for now, we will highlight a few salient points. First, while much of the literature places great importance on reducing social and economic inequalities, this framework suggests that such spatial variables are just one dimension of inclusion/exclusion among several. This reiterates our earlier cautionary note regarding a narrow focus on health inequality. Second, inclusion is underpinned by a set of psycho-social factors associated with a subjective sense of feeling fully part of a group or community. Belonging is much more than merely being present in the same space as others or knowing them. It connects with the universal human need to feel deeply connected to a group or community and is associated with psychological, physical, social, economic, and behavioural outcomes [21]. Third, included individuals do not just live within communities, but they also participate in them and contribute to them. This is important, as there is an implicit presumption in much of the literature on health in/equality that marginalised groups are primarily the recipients of health promotion or education. A properly inclusive approach, we suggest, demands that knowledge, skills, and talents are both developed and utilised so individuals can fully participate in the social and economic mainstream. Fourth, it is important to recognise the pivotal role of power in the realisation of inclusive practices. Community-based health-related activities in most developed countries target—or should target—groups from diverse social, cultural and linguistic backgrounds. Who determines the character and content of these activities? Or conversely, ‘Who has to shift?’ [22]. Traditionally, ‘experts’ and ‘professionals’ hold control of the delivery of health programmes, and participants are expected to comply and work within its frame of expectations. In less formal contexts, mainstream cultural values are translated into expectations in which marginalised groups must integrate. However, inclusion sees a negotiation and shifting of power in response to the experiences and expertise brought by all members of a group. These dimensions are partly driven by arguments for social justice, and the need to recognise and celebrate the diversity of abilities, needs and wants of all members of a community [23], but they also acknowledge the compelling real-world support for the principles of empowerment and self-determination showing that healthy lifestyles programmes are much more likely to deliver their objectives when all community members share a sense of belonging and ownership of those programmes.

## 2. Methods

Our first method for developing and presenting an account of inclusive healthy lifestyle promotion is by exploring some key distinctions in the ways in which inclusion has been discussed in the literature. Distinctions are vitally important in any educational context as they inform and direct what is perhaps the most fundamental educational question: ‘why do one thing rather than another?’ Distinctions can take different forms including those of content, delivery, and evaluation, but our interest, here, is with some distinctions between ideas and proposals.

Our second method is to examine the conceptual resources that are already available in the works of literature in support of girls/women, minority ethnic groups, and persons with disabilities, and to ask whether and to what extent these resources might help us to promote inclusive healthy lifestyles. By the end of this discussion, we hope to have shown that the case for a strong form of inclusion is warranted in healthy lifestyle promotion.

In seeking to explore these questions in a novel and, we hope, in a practicable way, to cast our conceptual net beyond public health. Specifically, we draw quite liberally in analyses if inclusion in cognate fields, such as education (especially physical education) and community sports participation.

## 3. Discussion

### 3.1. Integration and Inclusion: Some Lessons from Sports-Based Models

The concepts of integration and inclusion are frequently conflated. Take, as an example, the ‘Integration Continuum for Sport Participation’ [24]. We do not wish to suggest that sports performance and health-related physical activity are the same. However, sport is a sub-set of physical activity [25] and it is a context in which theorists have offered some useful distinctions that might inform more generalised discussions. Figure 1 shows a simple graphic representation of the integration continuum.

This model represents alternative ‘settings’ of sport for people with disabilities, ranging from regular sport with no modifications to segregated sport [24] (pp. 157–158). The settings are distinguished on the basis of the ‘degree of integration’ and ‘sport type’, and range from the least restrictive setting possible (1. Regular Sport), which is described as ‘most normal/integrated’, to the most restrictive segregated one (5. Adapted Sport Segregated). The other settings located between those two poles are: ‘2. Regular Sport with Accommodation’, which is described as necessarily “reasonable and [should] allow individuals with handicapping conditions equal opportunities to gain the same benefits or results from participation in a particular activity” [24] (p. 159); ‘3. Regular and Adapted Sport’; and ‘4. Adapted Sport Integrated’. The wording of this explanation has not aged well. However, we suggest that the basic approach will be familiar to many people working in community PA settings. That may be because Winnick’s framework was the starting point for the very widely used ‘Inclusion Spectrum’, initially presented in 1996 by Ken Black, and developed through various iterations [26]. Black changed the language of Winnick’s model, but more importantly, he removed the hierarchical design. Winnick assumed that participants should progress towards ‘regular sport’ and that the modifications he described were just stepping stones along the way towards that goal, but Black and his collaborators sought to arrange “the format of the continuum in a manner that gave each strategy equal importance” [27] (p. 123). The result is neither a continuum nor a spectrum (see Figure 2) (’STEP’ is the acronym for four key strategies that can be differentiated to promote participation—‘Space’, ‘Task’, ‘Equipment’ and ‘People’).

An interesting refinement was made to this model by locating ‘disability sport activity’ at the centre of the spectrum of settings. The proposal was that certain activities drawn from disability sport could be included in the other approaches, what the authors called ‘reverse integration activity’ [26]. An example of reverse integration might be including participants with and without impairments in a game of wheelchair basketball [28].

There are, of course, superficial similarities between these two models, but there are also significant differences. The progression from Winnick’s Integration Continuum to Black and colleagues’ Inclusion Spectrum is not merely a semantic change; it marks a step-change in practical approaches to difference and diversity. Black’s approach acknowledges that ensuring access does not, by itself, guarantee inclusion. Speaking from education, Barton [29] usefully draws out the central issue:


*“[Inclusion] is about responding to diversity; it is about listening to unfamiliar voices, being open, empowering all members and about celebrating ‘difference’ in dignified ways. From this perspective, the goal is not to leave anyone out … Inclusive experience is about learning to live with one another. This raises the question of what [an inclusive approach] is for. They must not—as was the case with many definitions of integration … be about assimilation in which a process of accommodation leaves [sport] remaining essentially unchanged.”*
(p. 234)

Winnick’s framework was a remarkable advance when it first appeared. However, it can be criticised with the benefit of hindsight for its presumption that the goal of integration is to aid disabled people into mainstream sports settings. Kiuppis [30] (p. 13) adds, “Sport appears here as a context, in which what is regarded as relevant is not the individual’s independent choice of a setting on the continuum from special to inclusive, but rather the replacement of special offers by integrative ones”.

The evolution of inclusive practices in sport helps to highlight a vital distinction between the integration of a relatively disempowered marginalised group into mainstream contexts and genuinely inclusive approaches that are about more than guaranteeing access. Examples from other groups’ integration/inclusion, such as those characterised by gender, race and ethnicity, class and age, would present intriguing alternative insights, but all cases would share a common assertion of the central importance of improving participation for all, irrespective of their personal or cultural characteristics. Kiuppis [30] frames this distinction in terms of moving beyond the question of ‘who’ (in the process-oriented sense) toward the ‘how’ of ensuring a range of opportunities for marginalised groups. This ‘who’/‘how’ distinction can be summarised in a simple table that also prefigures some of the discussions that are to follow. It uses physical activity to exemplify the points (see Table 1).

The transition from integration to inclusion is not just semantic, demanding, as it does, more far-reaching changes to wider sporting or physical activity systems. Evidence that an individual is integrated into a setting comes primarily from his or her physical presence, and further engagement is largely conditional on the ability and willingness to assimilate into the existing structures and systems of PA provision. Perhaps the clearest example of this phenomenon can be seen in girls and women wishing to join the so-called ‘male’ sports. Competitive sporting activities epitomise dominant forms of masculinity—or hegemonic masculinity—in terms of athleticism, heterosexuality, strength, ego and competence [32]. A substantial research literature piece has reported on the difficult “balancing act” many women face between being a sportsperson and being a feminine woman [33] (p. 709). The alternatives confronting many girls and women are either to acquiesce and assimilate or go somewhere else.

Compelling examples also come from studies of disabled people. For example, research with disabled members of Australian community sports clubs suggested that participants with disabilities were most likely to be accepted by a club when they did not require clubs or coaches to change their goals and practices significantly [34]. Kitchen and Howe [35] highlight similar issues in English cricket clubs, suggesting mainstreaming or integration implicitly imposed standards of entry based on (in these cases) abled-bodied norms and standards.

### 3.2. Exclusive Inclusion or Inclusive Inclusion

Most discussions of inclusion have referred to the treatment of specific populations. For example, the US Centre for Disease Control [36] explicitly explains ‘inclusive physical education and physical activity’ in terms of the participation, support, and encouragement of disabled students. Others frame ‘inclusive physical activity’ within the context of immigrants [37], members of LGBTQ+ communities [38], girls and women [39], and overweight people [40]. This narrative is not made distinctive by its focus on specific groups, *per se*, but by the absence of reference to other groups for whom the concept of inclusion could potentially apply. So, this approach can be called ‘exclusive inclusion’ because its focus is on specific groups and implicitly excludes those who fall outside its self-defined boundary.

In contrast, an ‘inclusive inclusion’ stance frames inclusion as an interdisciplinary and multi-factorial framework that prioritises fair and equitable access and participation of all citizens irrespective of differences [41]. It is called ‘inclusive inclusion’ as no group falls outside its boundary. Although this is an unusual view, it is not without precedent. DeLuca’s [42] article is a landmark contribution to holistic frameworks of inclusion, offering an overarching account rather than focusing on practice for particular groups of students who are categorised, labelled and targeted (such as those ‘with disabilities’ or those ‘from minority ethnic groups’). The framework helps to reveal the flawed nature of categorisation as a basis for thinking about inclusion [43] and offers an expansive account of inclusion better suited to identifying commonalities in people’s PA experiences. Artiles [41] argued that this approach could serve to both encourage research through a common conceptual framework and support developments in policy and practice. It also adds a counterpoint to theories of inclusion that replace one type of barrier (between specific marginalised groups and mainstream society) with another barrier (between different marginalised groups). It is also capable of dealing with the multiple identities and attributes that everybody brings to PA contexts. Each of us expresses a unique combination of gender, ethnicity, social class, sexual orientation and dis/ability, and the relevance of this mix varies considerably in different settings. The suggestion, here, is not that these individual differences are all the same. On the contrary, they manifest the diversity that is the lifeblood of inclusion. Rather, the intention is to “still the waters” [44] and to identify shared issues across a wide array of groups. An inclusive inclusion approach should help us to understand some of the recurring patterns of exclusion as the basis of identifying policy-making, management and delivery practices that facilitate the inclusion of all.

### 3.3. Participation and Inclusion

The switch away from a focus on specific aspects of exclusion and towards a holistic account might also be understood as a move from an essentially ‘negative’ form (e.g., “reducing exclusion from the cultures, curricula, and communities”, [45] (p. 7)) to a ‘positive’ emphasis on the promotion of participation (“Inclusion is the continuous process of increasing the presence, participation and achievements of all …”, [46], unpaged).

The role of participation within accounts of inclusion has been a matter of debate. The authors of the influential ‘Index for Inclusion’ [45] place participation at the centre of their account of inclusion in education. It is, in fact, in the subtitle of the document: “developing learning and participation in schools”. Speaking from a public planning perspective, however, [47] argue that inclusion and participation are, in fact, distinct concepts:

*“Participation practices entail efforts to increase public input oriented primarily to the content of programs and policies. Inclusion practices entail continuously creating a community involved in coproducing processes, policies, and programs for defining and addressing public issues.*”[47] (p. 272)

This is asserting an idiosyncratic view of inclusion, but their caution does warrant consideration in light of the frequent equation of inclusion and participation [45]. However, rather than being conceptually distinct, as seems to be suggested above, we would suggest that there is a conditional relationship between the two concepts and that participation is a necessary, but not sufficient condition of inclusion. In other words, participation must be present for an activity, club, or setting to be considered inclusive, but in itself, participation is not enough. Other factors must also be in place. These might include non-discriminatory attitudes and values, legal measures, and support mechanisms, which are also needed to facilitate inclusion. In addition, participation can and has been interpreted in many different ways, and these might well include characterisations of participation that fall well short of inclusion [48].

As with sport-based models, it has been commonplace to discuss and categorise participation using some sort of continuum, typically a ladder. Many of these frameworks take a lead from Arnstein’s [49] ‘ladder of participation’, which described a continuum of increasing stakeholder involvement, from passive dissemination of information (which she called ‘manipulation’) to active engagement (‘citizen control’). Numerous alternative terms have been suggested for the different rungs of this ladder. Some of the more influential models are summarised and compared in Figure 3 [49,50,51].

Despite their differences in content and emphasis, each of these frameworks highlights the need to differentiate between degrees of participation and the values associated with them. Some forms of participation—specifically, those placed in the lower rungs of the ladders—can perpetuate inequalities and amplify the exclusion of certain groups. Equating participation with a mere physical presence within a context of exclusionary values and practices can—inadvertently or not—halt more thorough-going attempts to create inclusive contexts [52]. Indeed, it has been suggested that many practitioners have found this aspect of the frameworks to be the most useful function of these models by helping them recognise and work to eliminate these forms of what we could call ‘pseudo-participation’ in their practice. Ironically, then, the greatest practical benefit of the creators of these participation frameworks may be in the exposure to these false types of participation, as much as his classification of the more positive types [53]. Hart’s “Ladder of Youth Participation” [50] is the most recognised framework and was initially designed as a modelling tool to describe possible levels of participation in the context of research projects. By figuratively ‘climbing’ the ladder, participation occurs with increased engagement and active involvement.

These ladder-based frameworks seem undeniably useful in articulating various youth participation types, but they are also limited when generalised to new contexts, such as those supportive of HEPA. Perhaps the most fundamental is the presumption of most of the framework authors that participation is hierarchical with citizen/child/community control as the ultimate goal [48]. Numerous healthy lifestyles promotion initiatives would seem dependent on the central involvement of knowledgeable providers and managers, and their displacement by citizens might hinder rather than support participants’ development, health, and empowerment. This concern is borne out by evaluations of youth–adult partnerships, in which activity quality and positive development outcomes were compromised when adults were not involved [54]. Likewise, health policy analysis suggests that citizen control of provision does not always align with the participants’ own reasons for engaging in decision-making processes, and the absence of such control can be unhelpfully interpreted as evidence of the failure or delegitimisation of otherwise valuable participatory processes [55].

A related criticism of ladders of participation is that they place an unfair burden of responsibility on participants [56]. It is well-established that factors such as socialising, fun and enjoyment, and skill learning are common and significant motivators for engagement in PA [57], and evidence in favour of leadership as a participatory goal are hard to find. Arnstein’s account presumes that roles and responsibilities change only in relation to changing levels of power (in the dynamic of citizens taking control and authorities relinquishing it). However, this obscures the complex set of relationships that exist in many ongoing situations where roles are less easy to define and responsibilities emerge during, and as a consequence of, the participatory process itself [58]. So, responsibilities are rarely prescribed and are much more likely to emerge over time, based on the individual construction of interest that would undermine hierarchical frameworks of participation.

Any attempts to bring about significant change to existing conditions in which forms of marginalisation will need to be identified, challenged and changed will necessitate an engagement with a host of issues. Actual participation is certainly necessary, but so also is power, and the distinction between these two concepts is well-captured by Hart [50]:

*“Young people’s participation cannot be discussed without considering power relations and the struggle for equal rights. It is important that all young people have the opportunity to learn to participate in programmes which directly affect their lives. This is especially so for disadvantaged children for through participation with others such children learn that to struggle against discrimination and repression, and to fight for their equal rights in solidarity with others is itself a fundamental democratic right … The highest possible degree of citizenship in my view is when we, children or adults, not only feel that we can initiate some change ourselves but when we also recognise that it is sometimes appropriate to also invite others to join us because of their own rights and because it affects them too, as fellow-citizens.*”(p. 8)

Hart’s discussion of power and participation raises the question of what public services and community provisions are for. It highlights issues of power and participation raises issues about the right to participate which, in turn, raises questions about the value of diversity and difference, and the recognition that to be excluded is to be disempowered, “to be constituted as ‘other’ and outside of a ‘normal’ frame of reference” [29] (p. 232). From this perspective, a foundational ambition of healthy lifestyles promotion programmes, including activities such as HEPA, is not to leave anyone out of fully engaging in those activities. This precludes methods of integration that are ultimately about assimilation in which a process of accommodation leaves the programme or setting remaining essentially unchanged.

### 3.4. From Normative to Transgressive Inclusion

One of the most nuanced frameworks for considering inclusion, both for us and other researchers in the field comes from Canada. DeLuca’s [42] holistic framework of inclusion is probably the most comprehensive attempt to date in clarifying thinking on inclusion. He proposes four conceptions and associated approaches to inclusion: normative; integrative; dialogical; and transgressive. Normative approaches enforce the assimilation and normalisation of marginalised individuals to the one dominant cultural standard while maintaining a dualistic discourse (i.e., dominant–subordinate/minority). They are acknowledged, but not legitimised, and cultural variations—language, dress, gender roles and so on—are largely overlooked and potentially altered [43]. This is a conditional acceptance: people from marginalised groups are welcome to join but joining depends on their acceptance of the dominant standard. Integrative approaches accept and legitimise the presence of difference in society by recognising the “duality between the dominant group and the minority group” [42] (p. 332), and respond to this difference through the use of differentiated activities to address individual needs. The frameworks proposed by Winnick [24] and, to some extent, Ken Black and his colleagues [25] align with this conception, although Black’s later work sought to undermine associations between participation and the achievement of fixed norms. This innovation is important as it indicates a movement towards a form of inclusion characterised by collaborative learning opportunities and personal relevance [43].

Dialogical approaches “bring forward knowledge as rooted in the lived, cultural experiences of diverse students” [42] (p. 334) by gathering ideas from different sources with the intention that all students will be enabled to participate fully in learning without prejudice. Whilst retaining a sense of a dominant group, dialogical interactions challenge assumptions and extend thinking, gathering ideas and practices from different sources. Diversity becomes a resource to tap, not a problem to solve [59]. With a transgressive approach to inclusion, individual diversity is “used as a vehicle for the generation of new knowledge and learning experiences” [42] (p. 334). There is no dominant cultural group, only overlays of divergent cultures that creates a shared and emergent learning. The word ‘transgressive’ is revealing as DeLuca’s account does not just seek to increase the extent of participation, but also change the character of that participation. It aims to increase awareness of how social stereotypes are seen as the norm, so that diversity may be “used as a vehicle for the generation of new knowledge and learning experiences” [42] (p. 334). This is only possible by weakening the hegemonic cultural dominance of one group, replacing it with a confluence of different cultures that create shared and emergent learning. Within the context of health, this approach suggests creating space for critical discussions and welcoming a clash of presumptions, which are surely necessary for advancing new ways of thinking about key topics. Within the context of education, Penney and her colleagues [43] connect these approaches with efforts to support students to question matters such as what it means to be ‘healthy’, ‘active’ or ‘fit’. This is made possible:

*“… through curriculum offerings, pedagogical approaches and assessment tasks that all align with this critical stance. Furthermore, the transgressive conceptualisation calls for curriculum that legitimises and prioritises exploration of the types of movement experience that are personally meaningful and rewarding to students.*”[43] (p. 1069)

Penney and her colleagues implicitly point to the importance of a sense of recognition and mutuality through PA experiences that “bring forward knowledge, as rooted in the lived, cultural experiences of diverse students” [42] (p. 334) and aim to encourage an understanding of participation away from the familiar and recognised toward diverse and new forms of practice that reflect an equitable and just society in action [60]. This highlights the limitations of labelling difference that emphasises a single issue or concern (disability, gender, and so on) or focuses on some type of difference and not others. DeLuca [42] calls this a “transgressive conception of inclusivity” (p. 334). Differences among participants are employed as vehicles for the creation of new learning experiences. Speaking from education, Dei and his colleagues [23] state that if teaching and learning include “the bodies, cultures, spaces, objects, positions, beliefs, sights, sounds, and smells within schools, then, an inclusive curriculum, which is positioned through the cultures and experiences of all students, is one that has the broadest range of academic possibilities” (p. 175). Likewise, the promotion of healthy lifestyles moves towards this transgressive conception of inclusion when it recognises the inherent diversity of bodies and minds of participants, their different wants and needs, and their unique personal biographies. This is only possible when all individuals are recognized as culturally complex within a shared power relationship. Cultural resources are overlayed, prompting emergent learning for all participants: “… learning cannot be standardized in this view because individual differences alter what and how learning takes place” [42] (p. 334). DeLuca describes the different accounts of the role of diverse cultures in terms of “hegemonic relationships” (p. 326). Traditional, or normative, approaches are ‘unicentric’ as they locate the dominant culture at the centre. They also maintaistrict dichotomy between dominant and subordinate groups that can magnify differences and undermine the worth of non-dominate groups’ contributions. So, marginalised groups are allowed to participate in activities on the condition that they conform to the expectations of the dominant group. Relationships become ‘multicentric’ when there is an acceptance of the presence of differences among social groups and individuals. However, while groups and individual differences are recognised, dominant cultural standards remain. DeLuca, Dei, Penney and their colleagues extend the conception of inclusion towards a ‘concentric’ orientation’: “there is no dominant cultural group, only overlays of cultures that create shared and emergent learning” (p. 334).

It might be noted in passing that there are parallels between transgressive accounts of inclusion and the philosophical theory of cosmopolitanism. Scheffler [61] distinguishes between two types of cosmopolitanism—‘cosmopolitanism about justice’ and ‘cosmopolitanism about culture’—which we can call political and cultural cosmopolitanism. Political cosmopolitanism opposes the idea that principles of justice regulate single societies with clear boundaries, arguing that these norms should be seen as applying to the global population as a whole. Cultural cosmopolitans challenge the idea that human flourishing depends on membership in distinct, stable cultural groups, defending instead our capacity to construct ways of life out of the heterogeneity of various cultural materials. The justice aspect of inclusion is widely acknowledged, but we suggest that a satisfactory account also requires a recognition of the inherent value of a diversity of cultural resources. In other words, inclusion demands a simultaneous recognition of the rights of all groups and individuals and an assertive recognition of the potential contribution of the ideas and practices offered by those groups and individuals. Thus, we suggest, inclusion is inseparable from recognition in both of these frames.

### 3.5. From Exclusion to Inclusion

As we have seen, accounts of inclusive practice have often been framed using spectra and continua. There is a strong intuitive appeal of this approach: sequential frameworks imply progressive movement towards a goal, rather than ‘Manichean’ thinking in which the scope of possibilities is reduced to simplistic dualities (light/dark; good/bad; inclusion/exclusion). Just as these dualities can offer useful points of reference in certain circumstances, so can talk of exclusion and inclusion. However, the reality is that change from one point to another occurs in gradually, covering several intermediate stages in the journey towards a goal. Or perhaps towards an unachievable ideal [62]?

Drawing on the above discussion, we attempted to summarise and represent some of the findings of the discussion in Figure 4 below:

To summarise the findings, we borrowed some of the language and iconography of some of the writers cited earlier in the article. For example, it implicitly reflects Dei et al.’s [23] discussion of hegemonic relationships between marginalised groups and dominant cultural groups, and the need to address inequality related to the distribution of power. It borrows from DeLuca [42] his rejection of hierarchical views that pit the marginalised against the *marginaliser* in favour of a more circulating understanding of hegemony that accepts intermediate and indeterminate relationships of power between cultures and culturally complex individuals. This position is reminiscent of Gramsci’s original notion of hegemony: power is a necessary part of natural social orders and is something that “circulates within a web of relationships in which we all participate, rather than as something imposed from top down” [63].

So, inclusion cannot be accommodated within a simple dualistic account: exclusion or inclusion; integration or inclusion. The sports-based models presented earlier in this article were effective in communicating this principle in terms of their ‘continuas’ and ‘spectra’, suggesting that inclusion and inclusive practice are directions of travel rather than destinations. This approach has an intuitive appeal as it brings the importance of continual action and reflection to the fore.

## 4. Conclusions

The purpose of this paper was to examine the concept of inclusion within the context of the promotion of healthy lifestyles, and specifically community-based HEPA and sport. The approach adopted in this task was to construct and question what we believe to be some central distinctions in the field, and in doing so to sketch some of the details of the necessary elements of a meaningful and defensible account of inclusion. As a human right, inclusion makes certain demands on those who are expected to accommodate it. This approach challenges what has been proposed as the central tension between in discussions of public health between the needs of societies and the demands of individual liberty. The concept of inclusion presented in this article is not reducible to one or the other as any account based on human rights simultaneously places expectations on societal agencies (such as those responsible for health promotion) and seeks to empower citizens’ liberty by removing barriers to their participation in activities related to promoting healthy lifestyles.

Our proposed framework for thinking about inclusion in terms of increasingly transgressive degrees of participation is offered tentatively and does not aspire to be a formal theory. Rather, it aims to prompt new thinking on inclusion in the context of HEPA and sport in terms of practices and ideas that address the needs of all participants, including those who have been traditionally marginalised within promoting healthy lifestyles.

## Figures and Tables

**Figure 1 ijerph-19-09917-f001:**
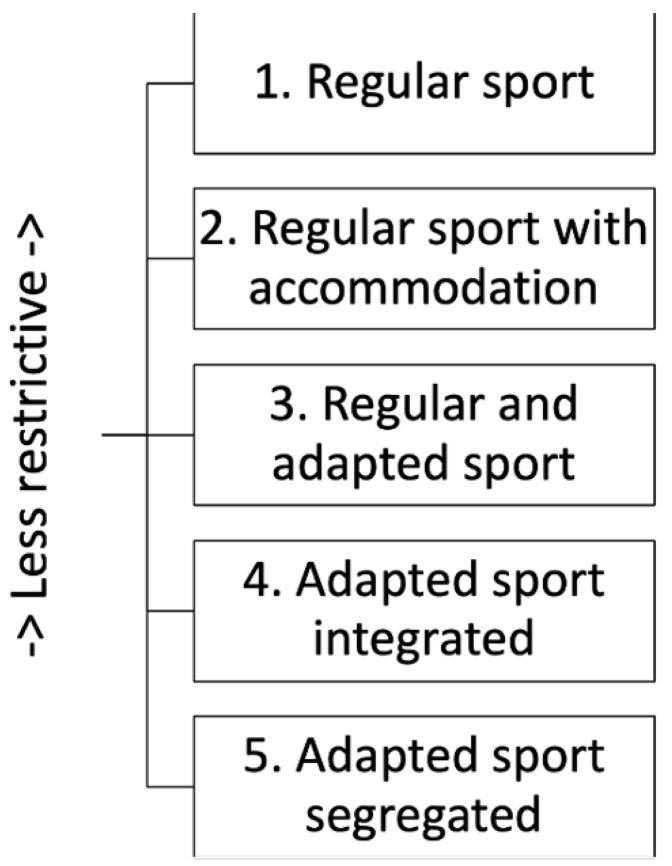
The integration continuum (based on [24]).

**Figure 2 ijerph-19-09917-f002:**
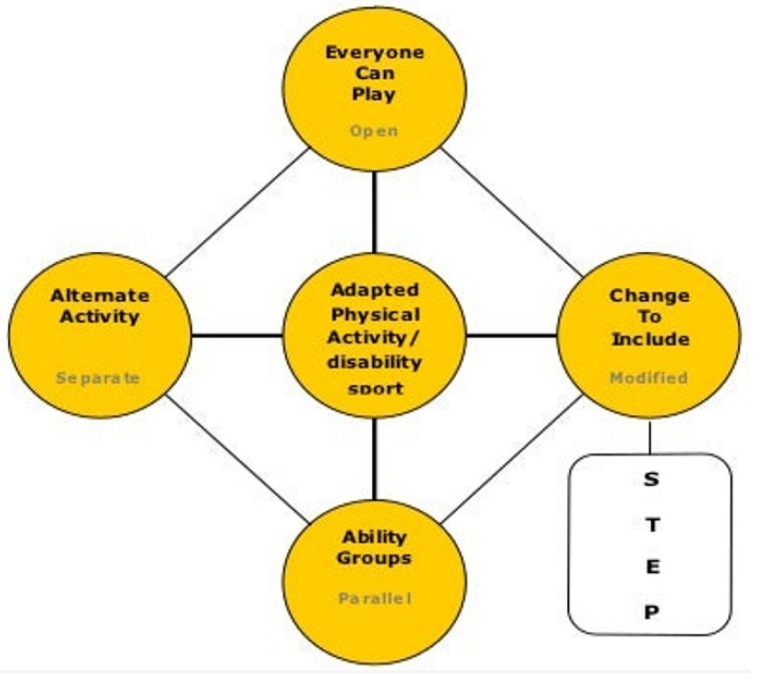
The inclusion spectrum (reproduced with permission from Ken Black).

**Figure 3 ijerph-19-09917-f003:**
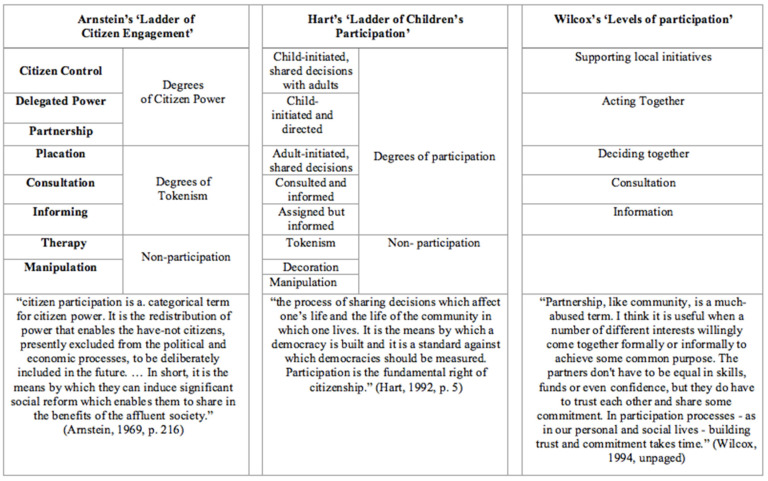
Models of Participation [49,50,51].

**Figure 4 ijerph-19-09917-f004:**
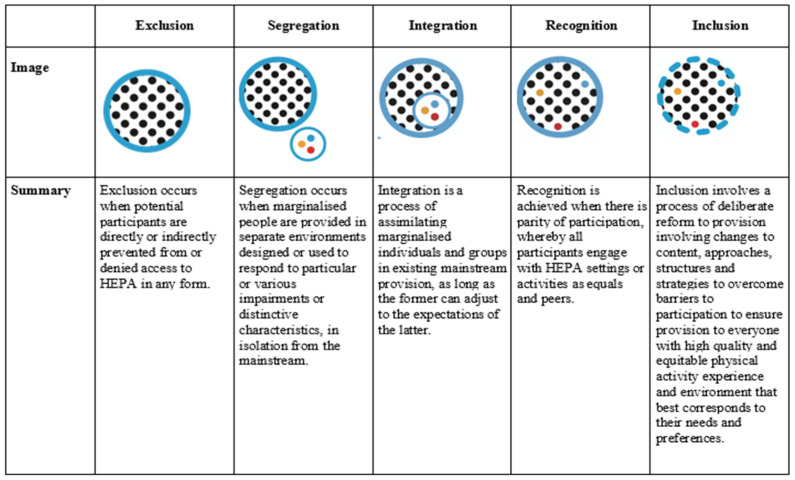
Graphical conceptualisation of forms of engagement in promoting healthy lifestyles.

**Table 1 ijerph-19-09917-t001:** Integration-based and inclusion-based approaches (loosely based on [31]).

Integration-Based Approaches	Inclusion-Based Approaches
Focus on individual’s needs (e.g., therapeutic exercise for specific impairments)	Focus on the rights of everyone (e.g., promoting ‘physical activity for all’ programmes)
Changing the individual (e.g., supporting individuals towards mainstream participation)	Changing the setting (e.g., adapting goals to be responsive to different groups)
Benefits to integrated individual	Benefits to everyone
Special programmes	Adaptive and supportive regular settings

## Data Availability

Not applicable.

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
