# Peer review of "Conceptualising Inclusion and Participation in the Promotion of Healthy Lifestyles"

_ijerph, 2022, doi:10.3390/ijerph19169917_

Round 1
Reviewer 1 Report
The article is very interesting, an important problem for the society has been discussed. The publications cited are aptly selected. Conclusions correctly drawn.
I have very few comments- rather technical:
Explain more clearly the survey methodology
Line 43: Maybe the latest data on physical activity after 2010 will be found
Line 568: 1916- a year of publication of the latest edition of this book?
Author Response
Responses To Reviewers’ Comments
Reviewer 1 Comments
(Explain more clearly the survey methodology)
- We are confused by this comment as there was no survey. Nevertheless, he has tried to make our methods clearer, in general.
(Line 43: Maybe the latest data on physical activity after 2010 will be found)
- Yes, this is a good point. Changed.
(Line 568: 1916- a year of publication of the latest edition of this book?)
- We have cited the issue of the book we consulted. We believe that giving dates of more recently re-issued books is only necessary when actually citing the re-issue.
Reviewer 2 Report
Thank you very much for the opportunity to review the article entitled “Conceptualising Inclusion and Participation in Healthy Life-2 styles Promotion”. As I understood, the article addresses the participation of individuals in practices associated with healthy lifestyles, mainly sport and physical activity in community settings. Several methods are used to explore different concepts and frameworks of integration, inclusion/exclusion, and participation of marginalised groups in sport and and community-based physical activity. The main result is a proposed framework that introduces a new thinking on inclusion in the context of health-enhancing physical activity and sport that addresses the needs of all participants including marginalised groups.
I find the topic of this article interesting and relevant. I also appreciate the time and energy both authors put into developing this article. However, I have several points regarding the structure and clarity of the article, which I will describe below.
General remarks
1. The introduction starts with a central tension in health promotion between a public policy that healthy lifestyles should be encouraged and liberal demands that states should not interfere with private affairs. To me it was not clear how this leads to the described aim of the article, namely the engagement of community members in local-level sporting and physical activities (PA) and associated clubs and organisations.
2. The term "healthy lifestyles" is used several times in the title, the introduction and methods section, and the produced framework aims to make sense of inclusion in the context of healthy lifestyle promotion. However, the term "healthy lifestyles" can have very different meanings depending on the underlying perspective (e.g. from those that emphasise individual health behavior change to those rooted in sociology emphasizing more structural factors). I think a brief explanation, or at least a source, would help readers understand how the term is used here.
3. I found the subheading sometimes confusing, especially 3.3. Participation and Inclusion were the corresponding section is very long. It starts with the relationship between participation and inclusion, goes to ladders of participation, then introduces conceptions and associated approaches to inclusion, introduces further the concept of “transgressive conception of inclusivity” and finally compares this with the philosophical theory of cosmopolitanism. I had to read and re-read this section several times, and found it difficult to keep up with all these different concepts, frameworks, theories etc. and why they were introduced. I think a slightly more descriptive subheading or to consider more subheadings could be helpful to readers.
4. There are different AIMS at different places in the article. I would recommend adjusting this. See:
- in this article, we wish to look at another aspect of health promotion, namely the engagement of community members in local-level sporting and physical activities (PA) and associated clubs and organisations (Line 33/34);
- So, this article aims to examine these issues and work towards some tentative suggestions for resolving them (Line 88/89),
- By the end of this discussion, we hope to have shown that the case for a strong form of inclusion is warranted in health promotion (Line 168/69);
- The purpose of this paper was to examine the concept of inclusion within the context of public health, and specifically community-based health-enhancing physical activity and sport (Line 510-512)
5. Several terms are used, but not in a consistent way. I suggest checking this and using as few different terms for one meaning as possible. I would also recommend checking whether a brief introduction of new terms would improve the readability of the article. See below:
- physical activity, inactivity, physical inactivity, sufficient moderate to vigorous physical activity, community physical activity, physical activity experiences, health-enhancing physical activity, health-related physical activity, community-based health-enhancing physical activity, local-level sporting, sport, competitive sporting activities
- Public health, public health promotion, health promotion, public health policy, public health programmes, public health initiatives
- within the context of healthy lifestyle promotion, within the public health context, within the context of health promotion, educational context, within the context of immigrants […], within a context of exclusionary values and practices, in the context of research projects, within the context of health, in the context of health-enhancing physical activity and sport
- community settings, specific settings, less formal settings, alternative ‘settings’ of sport, community physical activity settings, spectrum of settings, mainstream sports settings, mainstream settings, physical activity setting, different settings, inclusive settings, new settings, within public health settings, HEPA settings in figure 5
See further comments line by line:
· Line 46-49: Is this a relevant information here or could it be deleted?
· Lines 67-71 and 76/77: I found the sentences hard to understand. What do you mean by "[…] risk conflating symptoms with causes, and in doing so pay insufficient attention to the processes"? Why do you need the introduction of this framework about root causes of health inequalities for the introduction of the processes of inclusion/exclusion "as its intention was different from ours" as you mention in line 84?
· Line 75/76: It seem that something is missing in the sentence.
· Line 81: What do you mean by practices associated with healthy lifestyles?
· Line 87: What do you mean by specific settings?
· Line 89-91: Is this not part of the method section?
· Line 102: What do you mean by social structures?
· Line 159: Which principle do you mean? Empowerment and self-determination? If so, then these are two principles.
· Line 97: This section confuses me, although I appreciate the content! However, until line 96 inclusion plays a minor role. Instead, the focus seems to be more on participation. What is the function of this relatively long subsection about inclusion here? Is this a description of inclusion as starting point or does the critical discussion of central concepts and distinctions of inclusion start here? I would suggest to add an introductory sentence at the beginning of this subsection were you could briefly describe why this subsection is important and why the reader needs to know this here.
METHODS
· Line 163: Why do you describe the educational context here instead of the public health or health promotion context?
· Line 167: This seems to be the only time were you bring up ideas and proposals. What do you mean by that specifically?
· Line 168/69: The subsection (lines 98 to 159) has already convinced me that the case for a strong form of inclusion in health promotion is warranted. See my earlier comment in Line 97.
· Line 165: The concepts of Sport and Sport Participation are rarely introduced at this point. To me, sport and sport participation can be very different from the concept of health-enhancing physical activity with different consequences for inclusion and participation. I would suggest introducing both terms to the readers and to make always clear when and why you refer to sport or health-enhancing physical activity.
DISCUSSION
· Line 241: Which system do you mean?
· Line 243: What do you mean by structures and systems?
· Line 291: What do you mean by practices here?
· Line 258: This is a nice chapter! However, I was wondering why you refer relatively often to the educational context/ physical education in the article as this is different from sports and HEPA. It is ok to introduce inclusion frameworks used in education, but I think it would be helpful to make this clear earlier in the article, maybe in the methods section.
· Line 319: You might consider deleting "interpretations and application" as that makes the sentence longer and does not add much information.
· Line 336/37: You might consider to slightly adjust the sentence in "Ironically, then, the greatest practical benefit of these participation frameworks may be in the exposure to these false types of participation, as much as in the classification of the more positive types."
· 338/42: Is this really an important information here or could this be deleted?
· Line 344/45: For me, public health and health-enhancing physical activity are not settings. "In the area of" would fit better as I understand it.
· 348/49: To my knowledge, the ladder-based frameworks (e.g. at least Arnstein) do not suggest to displace providers and managers by citizen. Therefore, I do not understand this argument here.
· 350-56: It is a long sentence and hard to read. You might consider making two sentences out of one. You could start the second sentence in line 352 with "Health policy analysis…”.
· Line 362: I wonder if it would be possible to give a short example from the field of sports or HEPA to support the reader's understanding.
· 390/91. Unfortunately, I do not understand this sentence. What is the message?
· 465-479: This is interesting, but how does this contribute to the aim of the article to examine the concept of inclusion within the context of public health, and specifically community-based health-enhancing physical activity and sport or to explain the engagement of community members in local-level sporting and physical activities (PA) and associated clubs and organizations?
· Line 490: Figure 5, HEPA as an abbreviation has not been introduced. You could also describe the abbreviation under the figure. What do mean by HEPA settings in the colum Recognition?
· Line 498: “intermediate and indeterminate relationships” appears the first time here. Could this be introduced earlier in the article before it is used in a summary?
CONCLUSION
· Line 509: Conclusion - The article is entitled "Conceptualizing Inclusion and Participation in Healthy Lifestyles Promotion". In the conclusion, participation is lost. It seems that participation is no longer mentioned after the introduction of the concept of "“transgressive conception of inclusivity” in line 440.
Author Response
(The introduction starts with a central tension in health promotion between a public policy that healthy lifestyles should be encouraged and liberal demands that states should not interfere with private affairs.)
- We have attempted to make this clearer by adding a clearer account in the Introduction and by returning to the theme in the Conclusion.
(The term "healthy lifestyles" is used several times in the title, the introduction and methods section, and the produced framework aims to make sense of inclusion in the context of healthy lifestyle promotion.)
- We have added our working definitions to the text.
(I found the subheading sometimes confusing, especially 3.3. Participation and Inclusion were the corresponding section is very long.)
- The 3.4 sub-heading was missing. This has now been added. In addition, one section of the text was unclear, and has been corrected.
(There are different AIMS at different places in the article. I would recommend adjusting this.)
- We have to disagree a little with the reviewer. These statements all point to the main objective of the paper, namely, to examine competing accounts of inclusion, and suggest our own ‘strong’ position.
(Several terms are used, but not in a consistent way. I suggest checking this and using as few different terms for one meaning as possible. I would also recommend checking whether a brief introduction of new terms would improve the readability of the article.)
- Thank you for this valuable suggestion. We have sought to ensure our usage of key terms is clearer and more consistent.
(Line 46-49: Is this a relevant information here or could it be deleted?)
- We think it is relevant as it supports the case for the importance of PA for health.
(Lines 67-71 and 76/77: I found the sentences hard to understand …)
- We think the reviewer has a good point, and we have removed much of the text, and the table. However, we wish to retain the comparison of equality and inclusion, as it is important for our subsequent discussion. The point about symptoms and causes seems quite clear to us, but we have added a short extension to our explanation to aid understanding.
(What do you mean by "[…] risk conflating symptoms with causes, and in doing so pay insufficient attention to the processes"?)
- We think the reviewer has a good point, and we have removed much of the text, and the table. However, we wish to retain the comparison of equality and inclusion, as it is important for our subsequent discussion. The point about symptoms and causes seems quite clear to us, but we have added a short extension to our explanation to aid understanding.
(Line 75/76: It seem that something is missing in the sentence.)
- Yes: missing words! This has now been corrected.
(Line 81: What do you mean by practices associated with healthy lifestyles?)
- This is explained a little later in the paragraph.
(Line 87: What do you mean by specific settings?)
- This refers to the different aspects of healthy lifestyles. As mentioned by this reviewer elsewhere, these are not really settings. So, we have replaced that word with ‘context’. And we have left ‘settings’ to refer only to spaces of participation.
(Line 89-91: Is this not part of the method section?)
- No. It is a summary of the aim of the paper before moving to the main discussion.
(Line 102: What do you mean by social structures?)
- We have simplified the sentence to remove this issue.
(Line 159: Which principle do you mean? Empowerment and self determination? If so, then these are two principles.)
- The sentence adds nothing to the discussion, so we have deleted it.
(Line 97: This section confuses me, although I appreciate the content! However, until line 96 inclusion plays a minor role.)
- We have attempted to introduce the idea that inclusion is fundamentally about participation. As such, it is introductory framing of discussions of inclusion. We have followed the reviewer’s suggestion of adding a sentence to this section to (hopefully!) make it clearer. And with the benefit of hindsight, we agree that this helps articulate the starting point of our argument.
(Line 163: Why do you describe the educational context here instead of the public health or health promotion context?)
- This has been corrected.
(Line 167: This seems to be the only time were you bring up ideas and proposals. What do you mean by that specifically?)
- We have edited this sentence to be clearer.
(Line 168/69: The subsection (lines 98 to 159) has already convinced me that the case for a strong form of inclusion in health promotion is warranted.)
- Fair point: the sentence also interferes with the clarity of the following sentence. The sentence has been removed.
(Line 165: The concepts of Sport and Sport Participation are rarely introduced at this point.)
- We have added a brief explanation of the relationship (as we understand it) between sports participation and HEPA.
(Line 241: Which system do you mean?)
- This has been made clearer.
(Line 243: What do you mean by structures and systems?)
- This has been made clearer.
(Line 291: What do you mean by practices here?)
- This has been made clearer.
(the educational context/ physical education in the article as this is different from sports and HEPA.)
- This is a useful suggestion, and we have tried to be more explicit on this matter.
(Line 319: You might consider deleting "interpretations and application" as that makes the sentence longer and does not add much information.)
- Agreed
(Line 336/37: You might consider to slightly adjust the sentence in "Ironically, then, the greatest practical benefit of these participation frameworks may be in the exposure to these false types of participation, as much as in the classification of the more positive types.")
- We think this is an important point and prefer to retain it as it is.
(338/42: Is this really an important information here or could this be deleted?)
- We think these is useful background information about the rationale for frameworks of participation.
(Line 344/45: For me, public health and health-enhancing physical activity are not settings. "In the area of" would fit better as I understand it.)
- We accept that ‘setting’ is a poor term. ‘Context’ seems to us a better term, as it is more ‘inclusive’ of settings and fields. This has been changed.
(348/49: To my knowledge, the ladder-based frameworks (e.g. at least Arnstein) do not suggest to displace providers and managers by citizen. Therefore, I do not understand this argument here.)
- We disagree with this criticism. Several of the models explicitly talk about ‘handing over’ control to citizens. We have added a source in support of this claim.
(350-56: It is a long sentence and hard to read. You might consider making two sentences out of one. You could start the second sentence in line 352 with "Health policy analysis…”.)
- We have done as suggested.
(Line 362: I wonder if it would be possible to give a short example from the field of sports or HEPA to support the reader's understanding.)
- We hope that the existing explanation is clear enough for the reader.
(390/91. Unfortunately, I do not understand this sentence. What is the message?)
- This has now been edited.
(465-479: This is interesting, but how does this contribute to the aim of the article to examine the concept of inclusion within the context of public health, and specifically community based health-enhancing physical activity and sport or to explain the engagement of
community members in local-level sporting and physical activities (PA) and associated clubs and organizations?)
- The intention, here, is to offer a new way of thinking about the potentially difficult idea of transgressive inclusion. We think it works quite well.
(Line 490: Figure 5, HEPA as an abbreviation has not been introduced. You could also describe the abbreviation under the figure. What do mean by HEPA settings in the colum Recognition?)
- The abbreviation ‘HEPA’ has now been introduced earlier and made consistent. - (Line 498: “intermediate and indeterminate relationships” appears the first time here. Could this be introduced earlier in the article before it is used in a summary?)
- This is not part of the summary. It is included to help explain DeLuca’s rejection of hierarchical models of inclusion.
(Line 509: Conclusion - The article is entitled "Conceptualizing Inclusion and Participation in Healthy Lifestyles Promotion". In the conclusion, participation is lost. It seems that participation is no longer mentioned after the introduction of the concept of "“transgressive conception of inclusivity” in line 440.)
- We have added a comment that reinforces the relationship between inclusion and participation. - Thank you again to the reviewers
Round 2
Reviewer 2 Report
Dear Authors,
Thank you for addressing my points! It was my pleasure to read this revised version of your manuscript. It's much clearer to me now. It is a theoretically sound and rich contribution to the promotion of a healthy lifestyle with a focus of participation and inclusion. Congratulations!